

# Computational linguistics based text emotion analysis using enhanced beetle antenna search with deep learning during COVID-19 pandemic

Youseef Alotaibi[1], Arun Mozhi Selvi Sundarapandi[2], Subhashini P[3] and Surendran Rajendran[4]

[1] Department of Computer Science, College of Computer and Information Systems, Umm Al-Qura University, Makkah, Saudi Arabia
[2] Department of Computer Science and Engineering Holycross Engineering College, Thoothukudi, India
[3] Department of Information Technology, Vel Tech Multi Tech Dr. Rangarajan Dr.Sakunthala Engineering College, Chennai, India
[4] Department of Computer Science and Engineering, Saveetha School of Engineering, Saveetha Institute of Medical and Technical Sciences, Chennai, India

Corresponding author
Arun Mozhi Selvi Sundarapandi,
arunmozhiselvisundarapandi@gmail.com

## ABSTRACT

Computational intelligence and nature-inspired computing have changed the way biologically and linguistically driven computing paradigms are made. In the last few decades, they have been used more and more to solve optimisation problems in the real world. Computational linguistics has its roots in linguistics, but most of the studies being done today are led by computer scientists. Data-driven and machine-learning methods have become more popular than handwritten language rules, which shows this shift. This study uses a new method called Computational Linguistics-based mood Analysis using Enhanced Beetle Antenna Search with deep learning (CLSA-EBASDL) to tackle the important problem of mood analysis during the COVID-19 pandemic. We sought to determine how people felt about the COVID-19 pandemic by studying social media texts. The method is made up of three main steps. First, data pre-processing changes raw data into a shape that can be used. After that, word embedding is done using the 'bi-directional encoder representations of transformers (BERT) process. An attention-based bidirectional long short-term memory (ABiLSTM) network is at the heart of mood classification. The Enhanced Beetle Antenna Search (EBAS) method, in particular, fine-tunes hyperparameters so that the ABiLSTM model works at its best. Many tests show that the CLSA-EBASDL method works better than others. Comparative studies show that it works, making it the best method for analysing opinion during the COVID-19 pandemic.

## INTRODUCTION

In recent times, the domain of computational linguistics has changed with its special focus from linguistics to computer science. At present, processes in computational linguistics were

made not by linguistic discovery, but computational optimisation and study of statistical properties (*Costola et al., 2023*). At present, it can be almost impossible for linguists without a computer science background to participate in computational linguistics, however, quite commonplace for computer specialists with limited knowledge of linguistics to participate. This domain that was previously controlled by linguists is developed so steeped in specific technical languages which could not be available without equally specialised training.

The world has been dealing with the new COVID-19 virus, and positive cases of the illness have profoundly impacted society (*Qorib et al., 2023*). The outbreak's impact was so huge that has made a comparison with dreaded pandemics and epidemics such as the Black Death (a form of bubonic plague) or the Great Influenza occurring in the past. The panic over this outbreak has crossed all over the world and affects millions of individuals through infection or worry, disruption, sadness, disgust, stress, and fear (*Joloudari et al., 2023*). This virus strain, regarded as a newer one, has killed more than half a million individuals and breached borders around the globe.

Researchers across the world were investigating the problem from many perspectives. This research pursuit even adds evaluating how Artificial Intelligence (AI) can predict and explain any paradigms made by the new COVID-19 disease. Meanwhile, governments have applied several measures, namely, isolation and social distancing, to prevent the spreading of this disease (*Alqarni & Rahman, 2023*). For netizens, social networking site is becoming an important interface for sharing vital data and a powerful space for a little misrepresentation for several internet users across the globe. The epidemic has been the most trending and talked about regarding the importance of online ever since it was stated in the previous week of February 2020 (*Ainapure et al., 2023*). The growth of social networking sites' utility for articulating feelings and views by the public has constituted opportunities for examining sentiments regarding some prevalent and dominant discourse.

Computer technologies offer profound opportunities for fighting communicable disease outbreaks and have an extraordinary play, particularly in sentimental analysis (SA) for mass media; this significance is because of its marvellous role in analysing public opinion (*Singh et al., 2022*). Many researchers denoted that several pandemics and outbreaks can be controlled promptly when specialists regard mass media data. Thus, SA in studying epidemics, like COVID-19, was significantly related to recent events. The coronavirus disease remains a controversial global topic on social networking sites. Recently, ML has gained popularity in research and has been applied in various domains, like computer security, medical applications, and so on (*Nezhad & Deihimi, 2022*). The unexpected surge of society's dependency on social networking sites for information in constant to conventional news springs, and the amount of data offered, brings forth amplified attention to the usage of natural language processing (NLP) and AI techniques for helping text analytics (*Pascual-Ferrá, Alperstein & Barnett, 2022*). This data adds different social phenomena, like natural hazards, cultural dynamics, public health, and social trends, matters often deliberated, and thoughts articulated by the one who utilizes mass media (*Flores-Ruiz, Elizondo-Salto & Barroso-González, 2021*). NLP and its implementation in the analyses of mass media encountered a dramatic expansion. However the difficulties inferring a text's inherent meaning by utilizing NLP techniques were still difficult. It

was revealed that the recent NLP technologies were "vulnerable to adversarial texts" (*Chandra & Krishna, 2021*). Accordingly, evolving a thoughtful of the limitations of text classifier techniques, along with the related methods in machine learning (ML), becomes imperative.

This study develops a novel Computational Linguistics Sentiment Analysis using an Enhanced Beetle Antenna Search with Deep Learning (CLSA-EBASDL) model during the COVID-19 pandemic. In the presented CLSA-EBASDL technique, the initial stage of data preprocessing was carried out to convert the data into a compatible format. For the word embedding process, the bi-directional encoder representations of transformers (BERT) process is applied in this study. Next, an attention-based bidirectional long short-term memory (ABiLSTM) network is used for sentiment classification purposes. Lastly, the EBAS technique is exploited for optimal hyperparameter adjustment of the ABiLSTM model. A wide range of experiments have been conducted to demonstrate the enhanced performance of the CLSA-EBASDL technique.

The study question that has been identified is how to analyse sentiment in the context of the COVID-19 pandemic, especially in the huge amount of text on social media. To solve this problem, our main goal is to create the CLSA-EBASDL model, which combines Enhanced Beetle Antenna Search with deep learning techniques. Through thorough experiments and analyses, we also want to show that our method works better than others. We help with current problems and also add to the area of computational linguistics and sentiment analysis as a whole by doing this.

The following sections are organised as follows: 'Literature Review' presents a Literature Review, 'The Proposed Model' outlines the identified problem with the Proposed Model, 'Results and Discussion' provides the methodology including data pre-processing, word embedding, and sentiment classification, and presents the results of extensive experiments. Finally, 'Conclusion' concludes the paper by highlighting the superior performance of the CLSA-EBASDL approach compared to existing algorithms.

## LITERATURE REVIEW

*Alsayat (2022)* focused on the enhancement of the performance of sentiment classification with the help of a modified DL method. It has an advanced word embedding method and constitutes an LSTM network. This study's contributions are 2-fold. Firstly, the author found a powerful structure related to word embedding and an LSTM which will learn the contextual link between words and comprehend rare or unseen words in comparatively evolving circumstances like the COVID-19 pandemic by recognizing prefixes and suffixes from trained data. Secondly, the author utilizes and captures the important alterations in existing techniques by suggesting a hybrid ensemble method for *Waheeb, Khan & Shang (2022)* modelled denoising AutoEncoder for eliminating noise in data, and the attentional system for the fusion of features of ELM-AE with LSTM can be implemented for classifying SA.

In *Yin et al. (2022)*, the authors conducted a comprehensive review of the discussion relevant to the coronavirus vaccine on Twitter. Here, the hot research topic deliberated

by individuals and the respective emotional polarity from the viewpoint of vaccine brands and nations. The outcomes reveal that many individuals hoped for the effectiveness of vaccines and were anxious to take vaccines. Conversely, negative tweets have been linked to news reports of post-injection side effects, post-vaccination deaths, and vaccine shortages. On the whole, this work will use the famous NLP technology to mine individual opinions on the COVID-19 vaccination on social networking sites and accurately visualize and analyse them. *Naseem et al. (2021)* examine views relating to COVID-19 by concentrating on people who communicate and share mass on Twitter. The tweets were labelled into neutral, positive, and negative sentiment classes. The author examined the gathered tweets for classifying sentiment utilizing various sets of classifiers and features. The negative opinion serves a significant role in training public sentiment, for example, the author detected that people favoured lockdown earlier in the epidemic.

In *Mohammed et al. (2022)*, a novel technique can be devised for the automated sentimental classification of coronavirus tweets utilizing the Adaptive Neuro-Fuzzy Inference System (ANFIS) method. The complete procedure comprises sentiment analysis, data collection, word embedding, classification, and preprocessing. In *Nemes & Kiss (2021)*, the author looks at the sentiments of individuals and utilizes tweets to determine how people have connected to coronavirus disease for a given period. Such SA was amplified by extracting information and named entity detection to receive a more inclusive picture. The SA depends on the bi-directional encoder representations of transformers (BERT) technique that can be the basic measurement method for the comparisons. In *Kaur et al. (2021)*, a study of Twitter data was made by the R programming language. They gathered Twitter data related to hashtag keywords, which included new cases, coronavirus, deaths, and recovered. In this work, the author has devised a method termed Hybrid Heterogeneous SVM (H-SVM) executed the sentiment classification and categorized those neutral, positive, and negative sentimental scores.

*Zhang et al. (2023)* proposed an innovative approach for EEG classification based on self-training maximum classifier discrepancy. The objective is to identify samples from new subjects that surpass the capabilities of the extant source subjects. This is accomplished by the proposed method through the maximisation of discrepancies between the outputs of two classifiers. Furthermore, an unlabeled test dataset is employed by a self-training method to extract knowledge from the new subject, thereby minimising the domain gap. To augment feature representation, a 3D Cube is built, which integrates both frequency and spatial information extracted from EEG data into the input features of a convolutional neural network (CNN). The effectiveness of the suggested approach in addressing domain transfer obstacles is demonstrated through rigorous experiments performed on the SEED and SEED-IV datasets, which ultimately yield superior performance.

*Nie et al. (2023)* presents an original dataset comprising sensor data obtained from motor activity recordings of 23 unipolar and bipolar depressive patients and 32 healthy controls. The initial phase of the experiment presented certain difficulties, as evidenced by the incomplete completion of less than 70 epochs and a Cohen Kappa score below 0.1 resulting from an imbalance in class distribution. However, the results of the subsequent experiment were notably better. By the outcomes of UMAP dimensionality reduction, the

model attained an accuracy of 0.991 despite the extended duration of the training epochs. The ultimate experiment utilised neural networks and UMAP in a synergistic fashion, employing an assortment of machine-learning classification algorithms. Unsupervised machine learning dimensionality reduction without neural networks was investigated in the paper, resulting in marginally lower scores (QDA). Quadratic discriminant analysis (QDA) is an approach to reduce demensionality for classification tasks in unsupervised learning. The study advances knowledge regarding the complex correlation between depression and motor activity by integrating assessments of depressive symptoms by specialists. This demonstrates the capability of ubiquitous sensor data to evaluate mental health disorders.

*Lu et al. (2023)* proceeded to classify and investigate established attention mechanisms in visual question answering (VQA) tasks, while also recognising their constraints and delineating ongoing improvements. We anticipate that as attention mechanisms are further investigated with great care, VQA will advance in a direction that is more intelligent and centred around human needs. In *Liu et al. (2023a)*, X endeavours to fill this void by augmenting the multi-label K-nearest neighbours (MLkNN) classifier with information from adjacent sentences and the complete text of tweets, in addition to features present in individual sentences. The modified MLkNN enables iterative corrections for multi-label emotion classification, which substantially improves the speed and accuracy of Twitter's brief text emotion classification.

This study introduces a novel semi-automatic method for annotating internet texts with multiple labels, creating a multi-labeled corpus for subsequent algorithm training (*Liu et al., 2023b*). A label is assigned to the initial two main emotional tendencies on each tweet, which typically consists of multiple sentences, with each sentence being tagged with an emotional tendency and polarity. Selected emotional identifiers, data preprocessing, automated annotation *via* word matching and weight calculation, and manual correction for instances of multiple emotional tendencies comprise the semi-automatic annotation. The effectiveness of the proposed method is demonstrated through experiments conducted on the Sentiment140 Twitter corpus, which demonstrate that semi-automatic and manual annotations are consistent. The study establishes annotation specifications and generates a multi-labelled emotion corpus consisting of 6500 tweets to train advanced algorithms.

This article addresses the challenge of efficient similarity retrieval using Computed Tomography Image sequences (CTIS) in a resource-constrained mobile telemedicine network (MTN) (*Zhuang et al., 2022*). Establishing progressive distributed similarity retrieval is challenging due to the time-series characteristics of CTI sequences and the limitations of MTNs. In reply, the article presents an innovative strategy called the Dprs method, which is tailored to the MTN environment. Significantly, prior research on Dprs processing in MTNs is limited, which elevates the quality of this contribution. The Dprs method, as proposed, integrates four essential supporting techniques. To begin with, implementing a PCTI-based similarity measurement improves the precision of the retrieval procedure. Additionally, a lightweight approach to safeguarding privacy considers the requirement for secure management of data. Thirdly, a data distribution scheme based

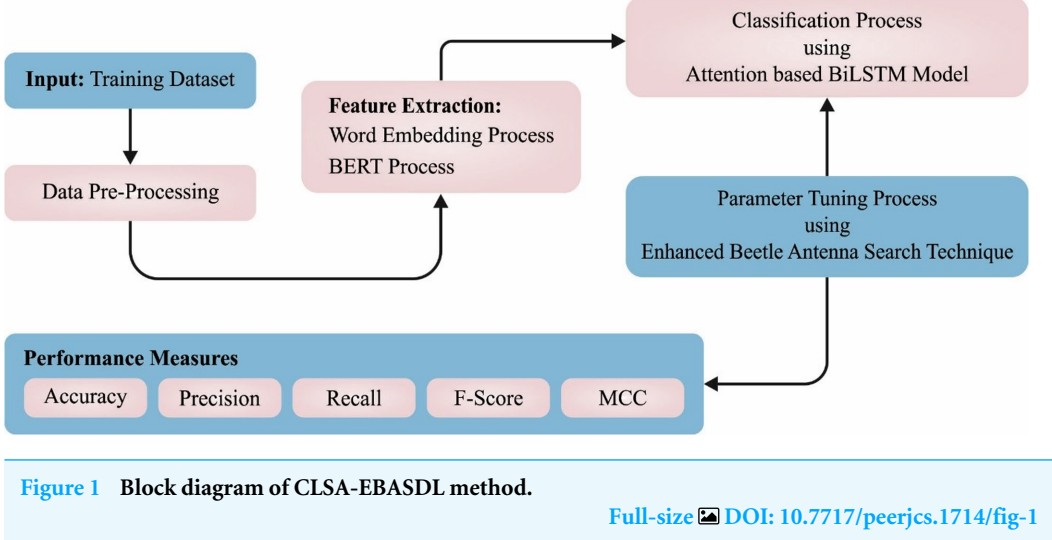

**Figure 1** **Block diagram of CLSA-EBASDL method.**

on SSL guarantees secure and efficient data transmission across the network. In conclusion, the UDI framework enhances the overall efficacy of the Dprs approach.

# THE PROPOSED MODEL

In this article, a novel CLSA-EBASDL method was introduced for the examination of sentiments that exist in the social networking media text during the COVID-19 pandemic. In the presented CLSA-EBASDL technique, the initial stage of data preprocessed was carried out for converting the data to a compatible format. For the word embedding process, the BERT process is applied in this study. Next, the ABiLSTM approach was used for sentiment classification purposes. Lastly, the EBAS method is exploited for optimum hyperparameter adjustment of the ABiLSTM network. Figure 1 depicts the working flow of the CLSA-EBASDL method.

## Data pre-processing

Primarily, the presented CLSA-EBASDL method performs data preprocessing to transform the information into a compatible format. Data analysis application requires data preprocessing to eliminate the redundant data for increasing the learning procedure of the classification model for improved performance. Redundant data describes any data without contribution or contributes minimum to forecasting the target class, but it raises the dimension of the feature vector and therefore presents superfluous computation difficulties. As a result, the accuracy of the classifier model is degraded if inappropriate or no preprocessing is performed. Therefore, data preprocessing or cleaning is carried out before encoding (*Thanarajan et al., 2023*). In this study, Python's NLP tool is utilized for preprocessing the tweeter dataset. Firstly, the texts are translated into lowercase, followed by link removal, punctuation, and HTML tags. Next, lemmatization and stemming models are carried out to clean the text, and finally, stopwords are detached.

- URL links, numbers, tags, and punctuation removal: This process does not contribute to enhancing the performance of the classification since they provide no further meaning for the learning model and raise the difficulty of feature space, thus assisting to decrease the feature space.
- Convert to lowercase: This process minimizes the difficulty of the feature subset as 'go' and 'Go' are considered distinct features in the ML model, thereby converting to lowercase would be 'go'. The model considers lower and uppercase words as dissimilar words that affect the training model and classification accuracy.
- Stopwords removal: Stop word is often utilized words that provide no beneficial information for analysis namely 'the', 'is', 'a', and 'an' are detached.
- Stemming and lemmatization: This process aims to reduce the derivationally correlated forms and inflectional forms of words into a base form. For instance, 'walks, 'walking', and 'walked' are transformed into the root word 'walk'.

## Word embedding

In this study, the BERT process is applied to the word embedding process. Word embedding depends on the distributed hypothesis of word representation. The word embedding characterizes natural language words as a lower-dimension vector representation that computers might understand (*Kumari, 2022*). The semantic relevance of words is evaluated by the comparison between vectors. At this point, it is widely employed in NLP tasks like BERT, Word2Vec, Glove, and so on. There exist two approaches to employing pre-trained language representation for downstream tasks: fine-tuning and feature-based models. Even though BERT is popularly applied in a fine-tuning model for NLP tasks, it can be utilised as a feature-based model and utilize the encoder for text representation. Similar to BERT, the WordPiece tokenizer has been utilized for the sequence of input text. The experiment shows that the WordPiece tokenizer is more efficient than the natural tokenizer which represents the technique of word segmentation depending on comma, space, and other punctuations, as well as the CoreNLP toolkit, which is widely employed in this study. BERT could express the tokenized words as the corresponding word embeddings, and the sentences are inputted as BERT modules, and the sentence vector representations of all sentences are attained.

$$r_m = BERT\,(s_m)\,s_m \in S, m \in [0, M].\qquad(1)$$

In Eq. (1), M denotes the sentence count, m indicates the index of the sentence, $s_m$ represents the text of the *m*th sentence, and S shows the group of sentences, $r_m$ indicates the sentence vector. Then, sentence vectors or word embeddings are utilized as input to the extractor and abstractor.

## Sentiment classification process

In this study, the ABiLSTM algorithm is used for sentiment classification purposes. It uses past datasets of sun irradiance from the target location as input features (*Wang et al., 2019*). The input can be signified by vector $X^T$, vector $Y^T$ shows the corresponding input data, and vector $Y^{T+\theta}$ characterizes the sentiment classification.

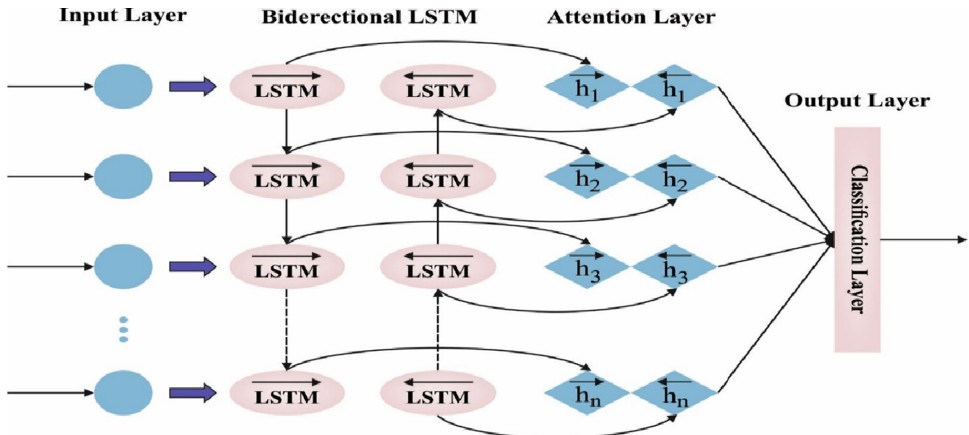

**Figure 2  Framework of BiLSTM.**

$$X^T = \left(X_{1'}X_2, X_3 \ldots X_T\right) \tag{2}$$

$$Y^T = \left(Y_{1'}Y_2, Y_3 \ldots Y_T\right). \tag{3}$$

Now, $T$ denotes the overall length of the time step and $\theta$ indicates the future time step. Since there exists no expressive dataset beforehand, the time window (w), and the input is set, $X(t-w), X(t-w+1), \ldots, X(t-1)$ are applied for calculating $Y(t+\theta)$ for all the tasks, in which $\Delta$ denotes the time frame in advance of prediction $Y$ indicates the prediction of input data utilizing a deep neural network $f$ on a formerly observed real-time dataset.

$$Y(t+\Delta) = f\left(X(t-w), X(t-w+1), \ldots, X(t-1)\right). \tag{4}$$

Previous datasets at $(t-w)$ time are signified as $I_{rr}(t-w)$, and the input parameter is characterized in the following expression for sentiment classification.

$$\{I_{rr}(t-w), I_{rr}(t-w+1), \ldots, I_{rr}(t-1)\}. \tag{5}$$

The features of autocorrelation and partial autocorrelation data were utilized for establishing the window size for the lag time sequence. The *i-th* hidden layer L, where the *i* value is fixed in model tuning, is characterized as $L_i$. In this study, the framework utilized for prediction is the attention-based BiLSTM-NN, which is composed: of encoder, attention, and softmax layers. Figure 2 demonstrates the infrastructure of BLSTM. The BLSTM acts as an encoding layer. The attention layer exploits the hidden output from these layers beforehand, creating a context vector to generate the scoring function. Subsequently, the classification of data performs decoding at the FC layer or transferred to the dense layer.

The recurrent neural network has been applied to model consecutive datasets in various problems. However due to problems with exploding or vanishing gradients, RNN is

not able to learn long-term dependency. To overcome this problem, the LSTM network is recommended and constructed based on RNN. Cell memory state and three gating mechanisms encompass an LSTM's architecture:

$$f_t = \sigma \left( W_f \left[ h_{t-1'} x_t \right] + b_f \right) \tag{6}$$

$$i_t = \sigma \left( W_i \left[ h_{t-1'} x_t \right] + b_i \right) \tag{7}$$

$$o_t = \sigma \left( W_o \left[ h_{t-1'} x_t \right] + b_o \right) \tag{8}$$

$$C_t = f_t * C_{t-1} + i_t * \tanh \left( W_c \left[ h_{t-1'} x_t \right] + b_c \right) \tag{9}$$

$$h_t = o_t * \tanh(C_t). \tag{10}$$

The LSTM cell biases $(b_j, b_f, b_o)$ and weighted matrices $(W_j, W_f, W_o)$ correspondingly represent the parameters of the input, forget, and output gates, The symbol $*$ denotes the sigmoid function and the element-wise multiplication. The word embedding of LSTM cell input is characterized as $x_t$ and the hidden state vector as $h_t$..

BiLSTM: The input is processed in sequential order using LSTM, which results in the effect of the previous input only and not the future one. The BiLSTM model was developed to make the model can also be influenced by future values. The LSTM processing chain is duplicated, which allows the input to manage reverse and forward time series, permitting the network to regard the future context of the network. The concluding output, $h_t$, of BiLSTM at the step $t$ is given below:

$$h_t = \left[ fh_t + bh_t \right]. \tag{11}$$

The attention module is used for considering the sensitive design variables. In real-time, the BiLSTM or LSTM will output a hidden $h_t$ layer at every time step.

The $h_t$ vector is intended into a single layer $MLp$ that learns hidden representations $u_t$. Next, $u_t$ and parameter context $uw$, a scalar significance value for $h_t$ is calculated. Lastly, the attention-based models exploit a *softmax* function to compute the weighted mean of state $h_t$ and it is shown below:

$$u_t = \tanh(W_w h_t + b_w) \tag{12}$$

$$a_t = \frac{e^{(u_t^T u_w)}}{\sum_t e^{(u_t^T u_w)}} \tag{13}$$

$$c = \sum_t a_t h_{t'}. \tag{14}$$

An FC softmax layer is applied as a classification. Vector $c$ is utilized as the feature for irradiation prediction:

$$\vec{y}_i = softmax(W_c C + b_c) \tag{15}$$

$\vec{y}_i$ shows the model prediction value, $W_c C$ indicates the weighted matrix, and $b_c$ denotes bias:

$$L = -\sum y_i \log \vec{y}_i. \tag{16}$$

In Eq. (16), $y_i$ indicates the observed irritation and $\vec{y}_i$ denotes the model predictive irradiation. The BP model derives the loss function for the whole set of parameters, and the SGD model is utilized for updating the model parameters.

### Hyperparameter tuning process

Lastly, the EBAS method is exploited for optimum hyperparameter adjustment of the ABiLSTM model. In BSA, we consider the search for food resources (viz., locations with a maximal intensity of food smell) as an optimization issue for mathematical modelling of the behaviour of a beetle (*Al Banna et al., 2021*). The maps of odor concentration in the atmosphere correspond to the value of the objective function. The aim is to search for the maximum odor concentration viz., the food resource. Consider $g(x)$ refers to the function demonstrating the odor concentration at point $x$. The maximal odor concentration corresponds to the solution of subsequent optimization issues using the linear dissimilarity constraint

$$\max_x g(x)$$

$$Subject \; to: x_{\min} \le x \le x_{\max}. \tag{17}$$

Assume at $t$ time instant, the beetle search at location $x_t$. The concentration of odor is represented as $(x_t)$, at the existing position. By taking the second phase, the beetle measures the concentration of odor in all directions through its antennae. Consider the antennae are positioned in opposite directions, and arbitrarily produced $\vec{b}$ characterizes the direction vector of the left antennae concerning the present location of beetle $x_t$.

$$x_l = x_t + \lambda \vec{b}$$

$$x_r = x_t - \lambda \vec{b}. \tag{18}$$

In Eq. (18), $\lambda$ indicates the length of the antennae, $x^l$, and $x^r$ denote the location vector of the left and right antennae correspondingly. But the vector might disturb the constraints of Eq. (17) due to arbitrarily made vector $\vec{b}$. Consequently, we determined a constraint set $\Psi$ in the following

$$\Psi = x | x_{min} < x < x_{max}$$

and projected the $x_l$ and $x_r$ vectors on the constraint set $\Psi$

$$\Psi_{X_l} = P_\Psi(x_l), \Psi_{\chi_r} = P_\Psi(x_r) \tag{19}$$

$\Psi$ denotes that the vector is anticipated on set $\Psi$. The $P_\Psi$ function is named a projection function.

$$P_\Psi(x) = \max\{x_{min}, \min\{x, x_{max}\}\}. \tag{20}$$

The explanation of the projected function is simpler and computationally effective. The odor intensity at the predictable antennae position is considered as $g(^\Psi x_l)$ and $g(^\Psi x_r)$. By relating the values, the beetle takes the second step based on the subsequent formula,

$$x_{new} = x_t + \delta sign\left(g(x_l) - g(x_r)\right)\vec{b}. \tag{21}$$

In Eq. (21), the signum function sign guarantees that the following phase is considered towards the direction of high odor concentration $\delta$ is the original step-size assumed by the beetle and proportionate to Euclidean distance among $x_{new}$ and $x_i$. Afterwards, accomplishing the novel position, the beetle re-measures the concentration of odor; when there exists an enhancement and it remains at the novel position; or else, it returns to the preceding position,

$$x_{t+1} = \begin{cases} x_{new'} & if, g(x_{new}) \geq g(xt) \\ x_{k'} & if, g(x_{new}) < g(x_t) \end{cases}. \tag{22}$$

Afterwards, accomplishing $x_{t+1}$, produce an arbitrary direction vector $\vec{b}$ and repeat the similar procedure until the attainment. Even though the abovementioned process is expressed for the maximized problem Eq. (17), it is transformed into the minimization problem by adapting the updating rule

$$x_{new} = x_t - \delta sign\left(g(x_l) - g(x_r)\right)\vec{b}. \tag{23}$$

The process is summarized below:
1. Begin from arbitrary position $x_0$.
2. Produce an arbitrary direction $\vec{b}$ vector for left antennae concerning the existing location $x_0$ of the beetle.
3. Compute the location of left and right antennae ($x^l$ and $x^r$) based on Eq. (18).
4. Estimate novel location $x_{t+1}$ based on Eqs. (21) and (22).
5. If obtained objective location $x_G$, stop. Or else, return to step 2.
   To resolve the optimization issue of targeted and untargeted attacks. We determine a matrix X

$$\overline{X} = [rcrgb]. \tag{24}$$

By assuming the notation, the element of $\overline{X}$ becomes $k \times 5$. This description of the novel matrix $\overline{X}$ allows for describing the objective function with one parameter.

$$g(\overline{X}) = \mathbb{P}(f_{cnn}(\mathfrak{P}(X_{img}, [\overline{X}[:,1], \overline{X}[:,2], X\{3,4,5\}])), C_{real}). \tag{25}$$

In Eq. (25), the semicolon symbol in the matrix indexing is utilized for representing the whole column of the matrix.

$$g(\overline{X}) = 1 - P\left(f_{cnn}\left(\mathfrak{P}\left(X_{img}, [\overline{X}[:,1], \overline{X}[:,2], \overline{X}\{3,4,5\}]\right)\right) C_{target}\right). \tag{26}$$

Here, $r$ and $c$ take integer values and then employ the round function for converting floating-point values to the integer.

To derive the EBAS technique, the Adaptive $\beta$-hill climbing concept is integrated into the BAS technique (*Zivkovic et al., 2021*; *Gowri, Surendran & Jabez, 2022*). To provide the present solution $X_i = (x_{i,1}, x_{i,2}, \ldots, x_{i,D})$, A $\beta$HC iteratively creates an improved solution $X_i'' = (x_{i,1}'', x_{i,2}'', \ldots, x_{i,D}'')$ on the beginning of 2 control operators such as $\mathfrak{N}$- and $\beta$-operators. The $N$-operator initial transmissions $X_i$ to a novel neighbourhood solution $X_i' = (x_{i,1}', x_{i,2}', \ldots, x_{i,D}')$ that is determined in Eqs. (27) and (28) as:

$$x_{i,j}' = x_{i,j} \pm U(0,1) \times N, j = 1, 2, \ldots, D \tag{27}$$

$$\mathfrak{N}(t) = 1 - \frac{t^{\frac{1}{K}}}{\text{Max}iter^{\frac{1}{K}}}. \tag{28}$$

In which $U(O,1)$ implies the randomly generated value within $[0,1]$, $x_{ij}$ defines the value of decision variables from the $j$th dimensional, $t$ represents the present iteration, *Maxiter* signifies the maximal iteration count, $N$ denotes the bandwidth distance amongst the present solution and their neighbour, $D$ stands for the spatial dimensional, and the $K$ parameter is a constant.

Then, the decision variable of novel solutions $X_i''$ was allocated both in the existing solution and the existing range of $\beta$-operator.

$$x_{i,j}' \leftarrow \begin{cases} x_{i,r}, & if r_8 < \beta \\ x_{i,j}', & else \end{cases} \tag{29}$$

$$\beta(t) = \beta_{\min} + (\beta_{\max} - \beta_{\min}) \times \frac{t}{\text{Max}iter}. \tag{30}$$

whereas $r_8$ refers to the random value between $[0,1]$, $x_{i,r}$ stands for another arbitrary number selected in the feasible range of that certain dimensional problem, $\beta_{\min}$ and $\beta_{\max}$ stands for the minimal and maximal values of probability values $\beta \in [0,1]$, correspondingly. When the created solution $X_i''$ is better than the present solution in consideration $X_i$, afterwards $X_i$ is exchanged by $X_i''$.

**Table 1  Dataset details.**

| Class | No. of samples |
| --- | --- |
| Positive | 7,663 |
| Negative | 1,768 |
| Neutral | 7,724 |
| Total Number of Samples | 17,155 |

The EBAS approach proceeds with FF to realize maximal classifier performance (*Motwakel et al., 2023*; *Alotaibi et al., 2021*). It resolves a positive integer to characterize the best efficacy of the candidate result. In this case, the diminished classifier error rate is examined as FF (*Mujahid et al., 2021*; *Rahman et al., 2023*). A good solution is a decreased error rate and the worst solution obtains a superior error rate (*Rahman et al., 2023*; *Mariselvam, Rajendran & Alotaibi, 2023*).

$$fitness(x_i) = Classifier\ Error\ Rate\,(x_i)$$

$$= \frac{No.\ of\ misclassified\ samples}{Total\ No.\ of\ samples} * 100. \tag{31}$$

# RESULTS AND DISCUSSION

The experimental analysis of the CLSA-EBASDL method is tested by the dataset including three classes (*Alshahrani et al., 2023*; *Al Ghamdi, 2023*; *Aldahlawi, Nourah & Sembawa, 2023*). The dataset contains 17,155 samples with three classes as given in Table 1 (The data is available at https://zenodo.org/records/10140889).

The confusion matrices exhibited by the CLSA-EBASDL method on varying training (TR) and testing (TS) databases are given in Fig. 3. With the 80%-TR database, the CLSA-EBASDL method has recognized 5,890 data instances with positive label, 1,386 data instances to negative label, and 6,025 data instances to neutral label. Likewise, with a 20%-TS database, the CLSA-EBASDL algorithm has detected 1,499 data instances with positive labels, 323 data instances with negative labels, and 1,500 data instances with neutral labels. Similarly, with 70%-TR data, the CLSA-EBASDL method has recognized 5,145 data instances with positive labels, 1,181 data instances with negative labels, and 5,257 data instances with neutral labels. Finally, with the 30%-TS database, the CLSA-EBASDL method has detected 2,253 data instances with positive labels, 537 data instances with negative labels, and 2,180 data instances with neutral labels.

Table 2 shows a brief sentiment classification outcome of the CLSA-EBASDL technique on 80%-TR data and 20%-TS database. Figure 4 exhibits the overall sentiment classification outcome of the CLSA-EBASDL technique on the 80%-TR database. The results implied that the CLSA-EBASDL method has reached better outcomes under all classes. For example, in positive class, the CLSA-EBASDL approach has offered $accu_y$ of 97.95%, $prec_n$ of 97.58%,

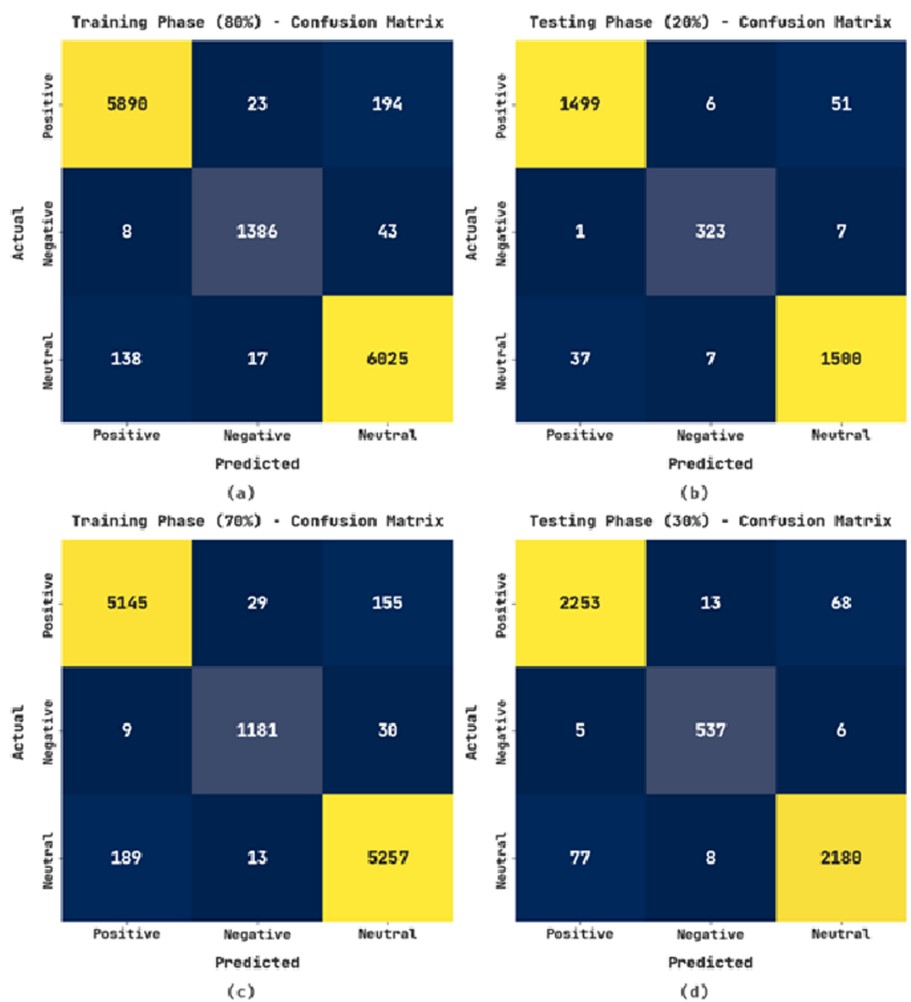

**Figure 3** Confusion matrix of CLSA-EBASDL approach (A and B) 80:20 of TR/TS data and (C and D) 70:30 of TR/TS data.

$reca_l$ of 96.45%, $F_{score}$ of 97.01%, and MCC of 94.64%. In the meantime, in the negative class, the CLSA-EBASDL algorithm has presented an $accu_y$ of 99.34%, $prec_n$ of 97.19%, $reca_l$ of 96.45%, $F_{score}$ of 96.82%, and MCC of 96.45%. In parallel, on the neutral class, the CLSA-EBASDL algorithm has rendered $accu_y$ of 97.14%, $prec_n$ of 96.22%, $reca_l$ of 97.49%, $F_{score}$ of 96.85%, and MCC of 94.24%.

Figure 5 displays the overall sentiment classification outcomes of the CLSA-EBASDL technique on the 20%-TS database. The outcome shows that the CLSA-EBASDL method has gained enhanced outcomes under all classes. For instance, on positive class, the CLSA-EBASDL method has rendered $accu_y$ of 97.23%, $prec_n$ of 97.53%, $reca_l$ of 96.34%, $F_{score}$ of 96.93%, and MCC of 94.41%. Simultaneously, for negative class, the CLSA-EBASDL approach has provided $accu_y$ of 99.39%, $prec_n$ of 96.13%, $reca_l$ of 97.58%, $F_{score}$ of 96.85%, and MCC of 96.52%. Eventually, in the neutral class, the CLSA-EBASDL technique has

**Table 2  Result analysis of CLSA-EBASDL method with dissimilar classes under 80:20 of TR/TS databases.**

| Labels | $Accu_y$ | $Prec_n$ | $Reca_l$ | $F_{score}$ | MCC |
|---|---|---|---|---|---|
| **Training Phase (80%)** | | | | | |
| Positive | 97.35 | 97.58 | 96.45 | 97.01 | 94.64 |
| Negative | 99.34 | 97.19 | 96.45 | 96.82 | 96.45 |
| Neutral | 97.14 | 96.22 | 97.49 | 96.85 | 94.24 |
| **Average** | 97.95 | 97.00 | 96.80 | 96.89 | 95.11 |
| **Testing Phase (20%)** | | | | | |
| Positive | 97.23 | 97.53 | 96.34 | 96.93 | 94.41 |
| Negative | 99.39 | 96.13 | 97.58 | 96.85 | 96.52 |
| Neutral | 97.03 | 96.28 | 97.15 | 96.71 | 94.00 |
| **Average** | 97.88 | 96.65 | 97.02 | 96.83 | 94.98 |

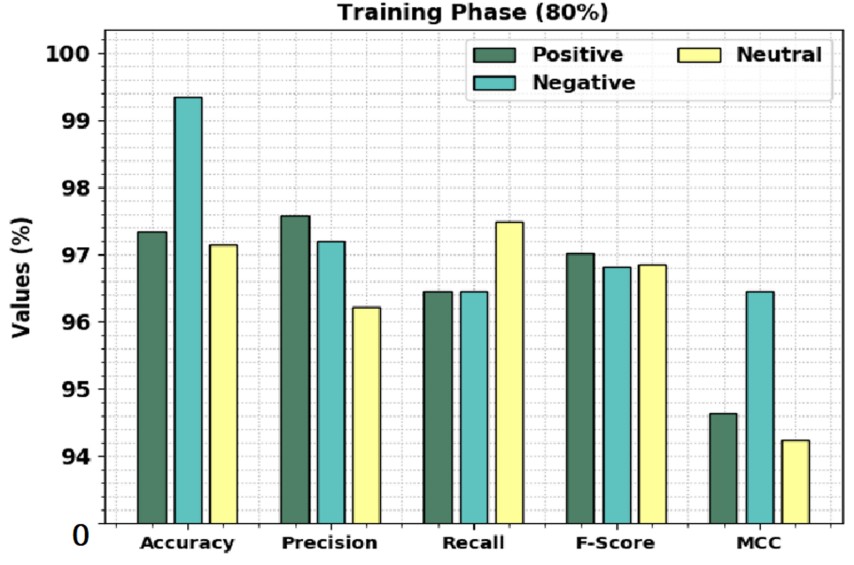

**Figure 4  Result analysis of CLSA-EBASDL approach under 80%-TR data.**

presented $accu_y$ of 97.03%, $prec_n$ of 96.28%, $reca_l$ of 97.15%, $F_{score}$ of 96.71%, and MCC of 94.00%.

Table 3 offers brief sentiment classification results of the CLSA-EBASDL technique on 70%-TR data and 30%-TS database. Figure 6 displays the overall sentiment classification results of the CLSA-EBASDL method on 70% of TR databases. The results denoted that the CLSA-EBASDL algorithm has reached enhanced outcomes under all class labels. For instance, in positive class, the CLSA-EBASDL method has offered $accu_y$ of 96.82%, $prec_n$ of 96.29%, $reca_l$ of 96.55%, $F_{score}$ of 96.42%, and MCC of 93.56%. Simultaneously, in the negative class, the CLSA-EBASDL method has presented $accu_y$ of 99.33%, $prec_n$ of 96.57%, $reca_l$ of 96.80%, $F_{score}$ of 96.68%, and MCC of 96.31%. Finally, in the neutral class, the

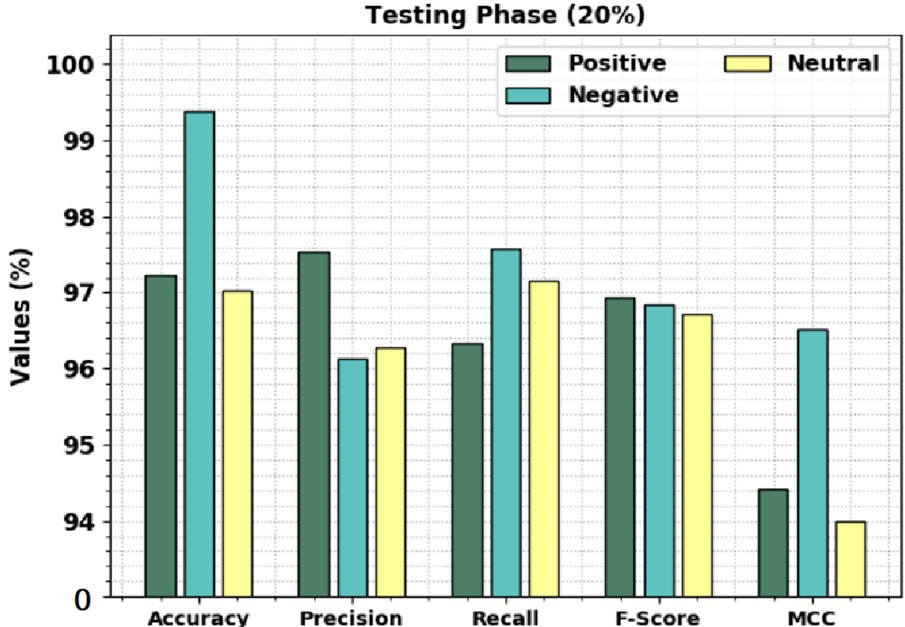

**Figure 5   Result analysis of CLSA-EBASDL method under 20%-TR database.**

**Table 3   Result analysis of CLSA-EBASDL method with dissimilar classes under 70:30 of TR/TS databases.**

| Labels | $Accu_y$ | $Prec_n$ | $Reca_l$ | $F_{score}$ | MCC |
|---|---|---|---|---|---|
| **Training Phase (70%)** | | | | | |
| Positive | 96.82 | 96.29 | 96.55 | 96.42 | 93.56 |
| Negative | 99.33 | 96.57 | 96.80 | 96.68 | 96.31 |
| Neutral | 96.78 | 96.60 | 96.30 | 96.45 | 93.50 |
| **Average** | 97.64 | 96.49 | 96.55 | 96.52 | 94.46 |
| **Testing Phase (30%)** | | | | | |
| Positive | 96.83 | 96.49 | 96.53 | 96.51 | 93.61 |
| Negative | 99.38 | 96.24 | 97.99 | 97.11 | 96.76 |
| Neutral | 96.91 | 96.72 | 96.25 | 96.48 | 93.73 |
| **Average** | 97.71 | 96.48 | 96.92 | 96.70 | 94.70 |

CLSA-EBASDL approach has granted $accu_y$ of 96.78%, $prec_n$ of 96.60%, $reca_l$ of 96.30%, $F_{score}$ of 96.45%, and MCC of 93.50%.

Table 3 offers brief sentiment classification results of the CLSA-EBASDL technique on 70%-TR data and 30%-TS database. Figure 6 displays the overall sentiment classification results of the CLSA-EBASDL method on 70% of TR databases. The results denoted that the CLSA-EBASDL algorithm has reached enhanced outcomes under all class labels. For instance, in positive class, the CLSA-EBASDL method has offered $accu_y$ of 96.82%, $prec_n$ of 96.29%, $reca_l$ of 96.55%, $F_{score}$ of 96.42%, and MCC of 93.56%. Simultaneously, in the negative class, the CLSA-EBASDL method has presented $accu_y$ of 99.33%, $prec_n$ of 96.57%,

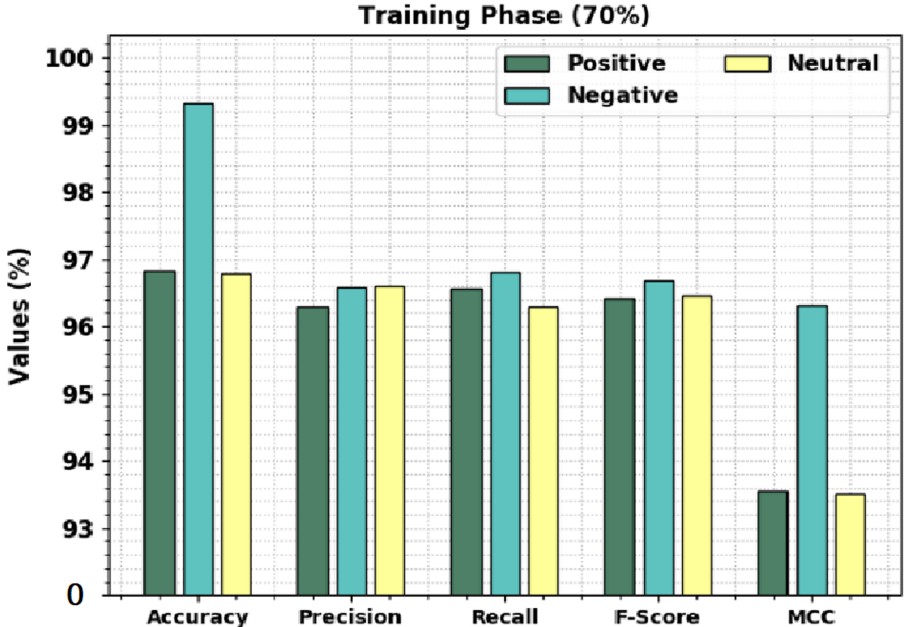

**Figure 6  Result analysis of CLSA-EBASDL method under 70% of TR databases.**

$reca_l$ of 96.80%, $F_{score}$ of 96.68%, and MCC of 96.31%. Finally, in the neutral class, the CLSA-EBASDL approach has granted $accu_y$ of 96.78%, $prec_n$ of 96.60%, $reca_l$ of 96.30%, $F_{score}$ of 96.45%, and MCC of 93.50%.

Figure 7 shows the overall sentiment classification outcomes of the CLSA-EBASDL approach on 30%-TS data. The results implicit in the CLSA-EBASDL technique have reached improved outcomes under all classes. For instance, in positive class, the CLSA-EBASDL method has provided $accu_y$ of 96.83%, $prec_n$ of 96.49%, $reca_l$ of 96.53%, $F_{score}$ of 96.51%, and MCC of 93.61%. In the meantime, for the negative class, the CLSA-EBASDL approach has presented an $accu_y$ of 99.38%, $prec_n$ of 96.24%, $reca_l$ of 97.99%, $F_{score}$ of 97.11%, and MCC of 96.76%. Eventually, in the neutral class, the CLSA-EBASDL method has rendered $accu_y$ of 96.91%, $prec_n$ of 96.72%, $reca_l$ of 96.25%, $F_{score}$ of 96.48%, and MCC of 93.73%.

A clear precision–recall inspection of the CLSA-EBASDL algorithm under the test database is shown in Fig. 8. The outcome implicit in the CLSA-EBASDL approach has resulted in improved values of precision–recall values under all classes.

A brief ROC study of the CLSA-EBASDL technique on test datasets is portrayed in Fig. 9. The results indicate that the CLSA-EBASDL algorithm has exhibited its capability in categorizing dissimilar class labels under test datasets.

Table 4 shows a comprehensive classification performance of the CLSA-EBASDL technique with the present approaches. Figure 10 provides a comparative study of the CLSA-EBASDL method with existing techniques in terms of $accu_y$. The outcome revealed that the ETC model has exhibited a reduced $accu_y$ of 79.31% whereas the RF and DT techniques have resulted in somewhat improved $accu_y$ of 85.84% and 83.21% respectively.

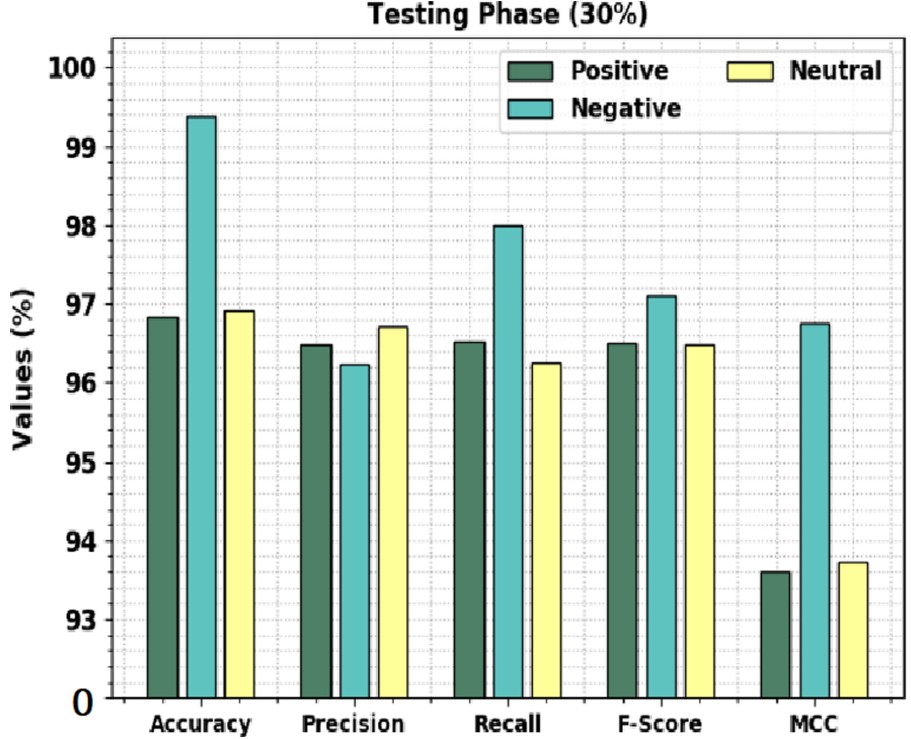

**Figure 7** Result analysis of CLSA-EBASDL method under 30%-TS database.

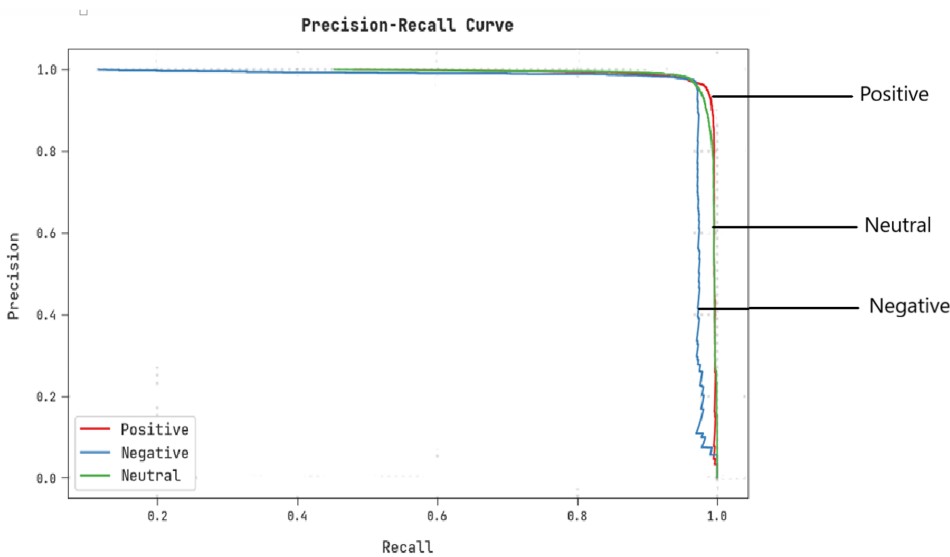

**Figure 8** Precision–recall examination of the CLSA-EBASDL method.

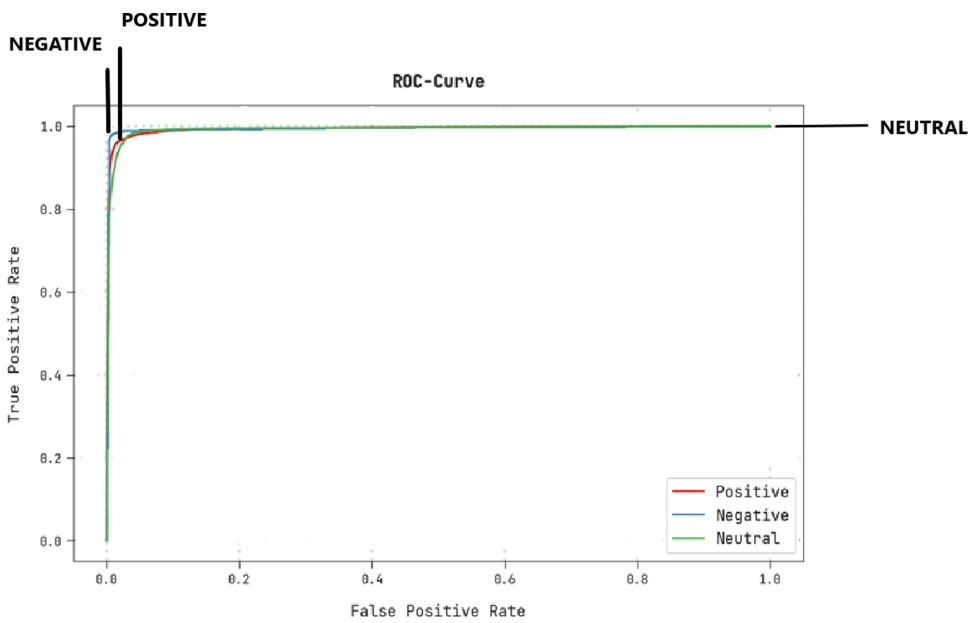

**Figure 9   ROC curve analysis of CLSA-EBASDL approach.**

**Table 4   Comparative analysis of CLSA-EBASDL method with existing methodologies.**

| Methods | $Accu_y$ | $Prec_n$ | $Reca_l$ | $F_{score}$ |
|---|---|---|---|---|
| CLSA-EBASDL | 97.88 | 96.65 | 97.02 | 96.83 |
| LR Model | 92.57 | 92.44 | 92.94 | 92.80 |
| RF Model | 85.84 | 86.52 | 85.61 | 86.70 |
| DT Model | 83.21 | 84.42 | 83.10 | 83.27 |
| SGD Model | 93.96 | 94.60 | 93.55 | 93.43 |
| SVM Model | 93.50 | 94.63 | 93.48 | 94.65 |
| ETC Model | 79.31 | 81.49 | 80.60 | 79.30 |

Then, the LR and SVM models reported reasonable outcomes with $accu_y$ of 92.57% and 93.50% respectively. However, the CLSA-EBASDL model has demonstrated superior outcomes with a maximum $accu_y$ of 97.88%.

Figure 11 offers a comparative study of the CLSA-EBASDL approach with recent methods in terms of $F_{score}$. The figure reveals that the ETC technique has displayed a reduced $F_{score}$ of 79.30% whereas the RF and DT algorithms have resulted in slightly improved $F_{score}$ of 86.70% and 83.27% respectively.

After that, the LR and SVM approaches have reported reasonable outcomes with a $F_{score}$ of 92.80% and 94.65% correspondingly (*Gowri, Surendran & Jabez, 2022*; *Motwakel et al., 2023*). However the CLSA-EBASDL technique has displayed enhanced results with a maximum $F_{score}$ of 96.83%. From the result analysis, it is clear that the CLSA-EBASDL model has offered enhanced sentiment classification outcomes.

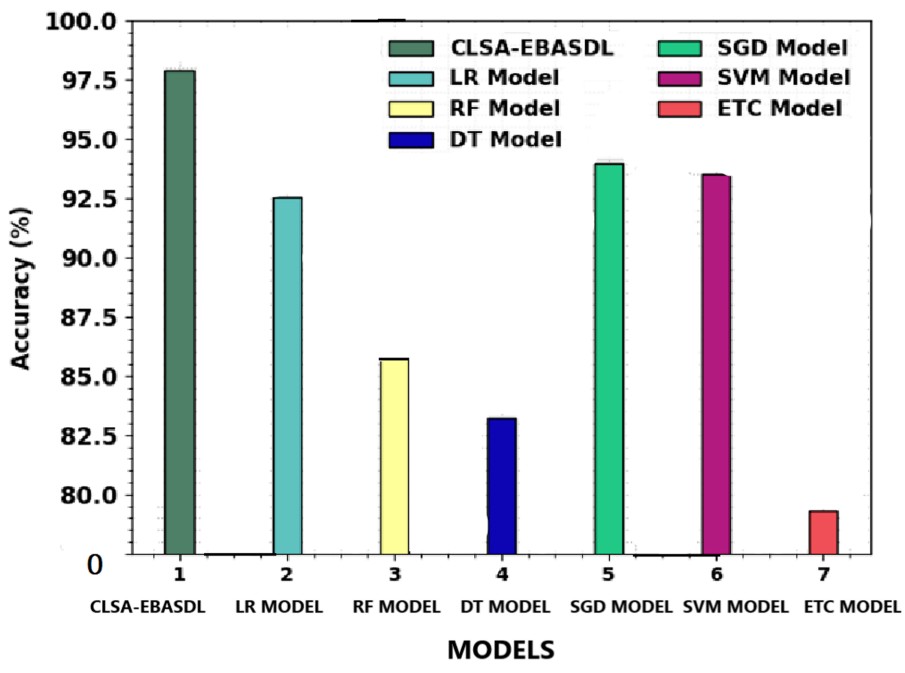

**Figure 10** Accuy analysis of CLSA-EBASDL methodology with existing approaches.

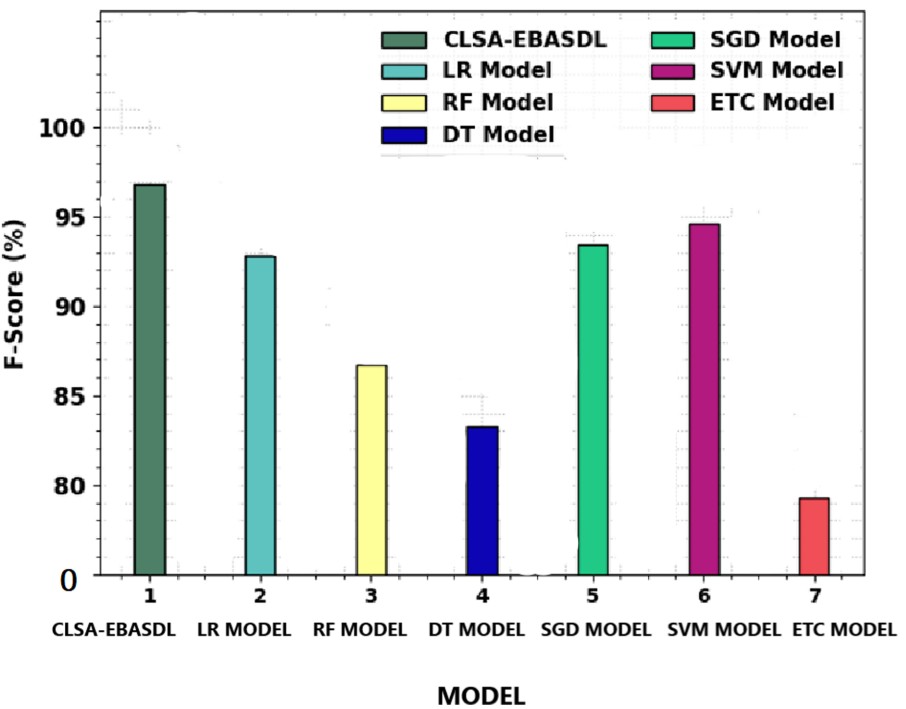

**Figure 11** F-score analysis of CLSA-EBASDL method with existing techniques.

# CONCLUSION

In this article, a novel CLSA-EBASDL system was introduced for the examination of sentiments that exist in the social media text during the COVID-19 pandemic. During the presented CLSA-EBASDL technique, the initial stage of data preprocessed was carried out for converting the data to a compatible format. For the word embedding process, the BERT process is applied in this study. Next, the ABiLSTM technique was used for sentiment classification purposes. Lastly, the EBAS technique is exploited for optimum hyperparameter adjustment of the ABiLSTM model. A wide range of experiments are applied to demonstrate the improved performance of the CLSA-EBASDL technique. The CLSA-EBASDL technique has displayed enhanced results with a maximum $F_{score}$ of 96.83%. From the result analysis, it is clear that the CLSA-EBASDL model has offered enhanced sentiment classification outcomes. The comprehensive analysis highlighted the supremacy of the CLSA-EBASDL system over existing algorithms. In the future, a DL based ensemble fusion algorithm will be developed to boost the overall sentiment classification performance of the CLSA-EBASDL technique. It is necessary to do further research to examine their sensitivity and resilience.

## Funding
The Deanship for Research & Innovation, Ministry of Education in Saudi Arabia, funded this research work through project number: IFP22UQU4281768DSR120. The funders had no role in study design, data collection and analysis, decision to publish, or preparation of the manuscript.

## Grant Disclosures
The following grant information was disclosed by the authors:
Research & Innovation, Ministry of Education in Saudi Arabia: IFP22UQU4281768DSR120.

## Competing Interests
The authors declare there are no competing interests.

## Author Contributions

- Youseef Alotaibi conceived and designed the experiments, prepared figures and/or tables, authored or reviewed drafts of the article, conceptualization, validation, formal analysis, funding acquisition, visualization, writing—review and editing, resources,, and approved the final draft.
- Arun Mozhi Selvi Sundarapandi performed the experiments, performed the computation work, prepared figures and/or tables, conceptualization, validation, supervision, and approved the final draft.
- Subhashini P conceived and designed the experiments, performed the computation work, prepared figures and/or tables, authored or reviewed drafts of the article, software, data curation, and approved the final draft.

- Surendran Rajendran performed the experiments, analyzed the data, authored or reviewed drafts of the article, methodology, validation, investigation, writing—original draft preparation, project administration, and approved the final draft.

## Data Availability

The data and code is available in the Supplementary Files.

The code is available at Zenodo: surendran, R. (2023). Computational linguistics based text emotion analysis using enhanced beetle antenna search with deep learning during COVID-19 pandemic. Zenodo. https://doi.org/10.5281/zenodo.10065580.

## Supplemental Information

Supplemental information for this article can be found online at http://dx.doi.org/10.7717/peerj-cs.1714#supplemental-information.

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
