# Peer review of "Computational linguistics based text emotion analysis using enhanced beetle antenna search with deep learning during COVID-19 pandemic"

_PeerJ Computer Science, doi:10.7717/peerj-cs.1714_

## Round 0.1 · original submission · Major Revisions

Dear authors,

Your article has not been recommended for publication in its current form. However, we encourage you to address the concerns and criticisms of the reviewers, particularly in terms of readability, originality and general quality, and to resubmit your article once you have updated it accordingly.

Best wishes,

**Language Note:** The Academic Editor has identified that the English language must be improved. PeerJ can provide language editing services - please contact us at copyediting@peerj.com for pricing (be sure to provide your manuscript number and title). Alternatively, you should make your own arrangements to improve the language quality and provide details in your response letter. – PeerJ Staff

Reviewer 1 ·

Basic reporting

1. The English is clear and unambiguous but there should be a review on the grammar.
2. The abstract should be structured in such a way to clearly state the following, (brief background, identified problem, objective, method used, result). The objective and the problem is missing.
3. I suggest that the contribution of the work be mentioned in the introduction section, and the introduction concluded with highlighting the other sections.
4. More literature is needed to provide sufficient background to the study and context.
5. Acronyms should be defined before use. Line 153 'BERT'. State the full meaning of the acronym.

Experimental design

1. The research work is within the aims and scope of the journal.
2. The problem statement should be clearly stated

Validity of the findings

The conclusion should summarize the findings with facts form the research work. Identify include the specific results that makes the work better when compared with similar research within the domain.

Reviewer 2 ·

Basic reporting

The paper develops a novel Computational Linguistics Sentiment Analysis using an Enhanced Beetle Antenna Search with Deep Learning (CLSA-EBASDL) model during the COVID-19 pandemic. The presented CLSA-EBASDL technique examines the sentiments of the social media text during the COVID-19 pandemic. In the proposed CLSA-EBASDL technique, the initial data pre-processing stage was completed to convert the data into a compatible format.

• The background of the study in the abstract section should be shortened; too lengthen. The authors should follow this structure for their abstract to be written for better understanding—background of the study, problem statement, objectives, materials and methods, results, conclusion and recommendation. Let the abstract be quantitative.
• The research issue and objectives should be highlighted in a separate paragraph in the introduction section (second to the last paragraph)
• The authors should summarize the remaining part of the article in the last paragraph.
• In section two alone, the in-text citation should be “In Alsayat [11]……; In Waheeb, Khan and Shang [12] ….. etc.
• The authors should highlight the contribution of their work by clearly identifying their contribution with particular respect to similar existing technical literature.
• What is the proportion of the data split
• Summarize the hyperparameter and values in a table
• The dataset details and all about the data descriptions should be moved to section three, and a new section, named “Data Description”, presents everything about the dataset used for this study, the repository from, the split, attributes, etc.
• It is essential to acknowledge the limitations of the study and engage in a discussion on potential avenues for future research.
• The authors should state how the study performance was accessed or evaluated.
• The study should be compared with existing systems (state-of-the-art), and authors should state how it surpassed the existing one and why it performed less or less.
• There is a need for significant improvement in the quality of the English language and how it is presented. The text had a notable quantity of typographical errors and grammatical issues. To ensure the quality of your paper, getting assistance from a colleague who is proficient in English and knowledgeable in the subject matter is recommended. Alternatively, you may consider contacting a professional editing service for their expertise.
• I request you kindly cite these suggested papers to provide a comprehensive perspective on your study. Citing these papers will enrich the scholarly context of your research and
a. Zhang, X., Huang, D., Li, H., Zhang, Y., Xia, Y.,... Liu, J. (2023). Self-training maximum classifier discrepancy for EEG emotion recognition. CAAI Transactions on Intelligence Technology. doi: https://doi.org/10.1049/cit2.12174
b. Nie, W., Bao, Y., Zhao, Y., & Liu, A. (2023). Long Dialogue Emotion Detection Based on Commonsense Knowledge Graph Guidance. IEEE Transactions on Multimedia. doi: 10.1109/TMM.2023.3267295
c. Lu, S., Liu, M., Yin, L., Yin, Z., Liu, X., Zheng, W.,... Kong, X. (2023). The multi-modal fusion in visual question answering: a review of attention mechanisms. PeerJ Computer Science, 9, e1400. doi: 10.7717/peerj-cs.1400
d. Liu, X., Shi, T., Zhou, G., Liu, M., Yin, Z., Yin, L.,... Zheng, W. (2023). Emotion classification for short texts: an improved multi-label method. Humanities and Social Sciences Communications, 10(1), 306. doi: 10.1057/s41599-023-01816-6
e. Liu, X., Zhou, G., Kong, M., Yin, Z., Li, X., Yin, L.,... Zheng, W. (2023). Developing Multi-Labelled Corpus of Twitter Short Texts: A Semi-Automatic Method. Systems, 11(8), 390. doi: 10.3390/systems11080390
f. Zhuang, Y., Jiang, N., Xu, Y., Xiangjie, K., & Kong, X. (2022). Progressive Distributed and Parallel Similarity Retrieval of Large CT Image Sequences in Mobile Telemedicine Networks. Wireless communications and mobile computing, 2022. doi: 10.1155/2022/6458350

Experimental design

Included in the basic comment section already

Validity of the findings

Included in the basic comment section already

Additional comments

None

---

## Round 0.2 · accepted · Accept

Dear authors,

Thank you for clearly addressing all of the reviewers' comments. Your article is accepted for publication after the revision.

Best wishes,

Reviewer 1 ·

Basic reporting

Clear and unambiguous English was used

The last paragraph of the introduction section has been amended as suggested.

More literature have been added as suggested

Experimental design

The research is within the aim and scope of the journal

Validity of the findings

The findings shows novelty

Additional comments

The work is better and should be accepted for publication

Reviewer 2 ·

Basic reporting

After carefully reviewing the manuscript and the author's responses, I am happy with the improvements made. The article has excellent language, a clear presentation, and meaningful results directly related to the study objective. After considering it, I can say that it deserves to be published. I appreciate the author's hard work in polishing the article; I believe it will add significantly to the current body of research.

Experimental design

No comment

Validity of the findings

No comment

Additional comments

No comment